

**PeerJ Hubs**
Published on behalf of

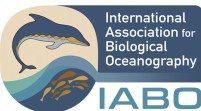
International Association for Biological Oceanography
IABO

# Comparison of feeding preferences of herbivorous fishes and the sea urchin *Diadema antillarum* in Little Cayman

Lindsay Spiers[1,2] and Thomas K. Frazer[1,3]

[1] Fisheries and Aquatic Sciences Program, School of Forest, Fisheries, and Geomatics Sciences, University of Florida, Gainesville, FL, United States of America
[2] Fish and Wildlife Research Institute, Florida Fish and Wildlife Conservation Commission, Marathon, FL, United States of America
[3] College of Marine Science, University of South Florida, St. Petersburg, FL, United States of America

Corresponding author
Lindsay Spiers, lspiers@elon.edu, Lindsay.Spiers@MyFWC.com

## ABSTRACT

On Caribbean coral reefs, losses of two key groups of grazers, herbivorous fishes and *Diadema antillarum*, coincided with dramatic increases in macroalgae, which have contributed to decreases in the resilience of these coral reefs and continued low coral cover. In some locations, herbivorous reef fishes and *D. antillarum* populations have begun to recover, and reductions in macroalgal cover and abundance have followed. Harder to determine, and perhaps more important, are the combined grazing effects of herbivorous fishes and *D. antillarum* on the structure of macroalgal communities. Surprisingly few studies have examined the feeding preferences of *D. antillarum* for different macroalgal species, and there have been even fewer comparative studies between these different herbivore types. Accordingly, a series of in-situ and ex-situ feeding assays involving herbivorous fishes and *D. antillarum* were used to examine feeding preferences. Ten macrophytes representing palatable and chemically and/or structurally defended species were used in these assays, including nine macroalgae, and one seagrass. All species were eaten by at least one of the herbivores tested, although consumption varied greatly. All herbivores consumed significant portions of two red algae species while avoiding *Halimeda tuna*, which has both chemical and structural defenses. Herbivorous fishes mostly avoided chemically defended species while *D. antillarum* consumed less of the structurally defended algae. These results suggest complementarity and redundancy in feeding by these different types of herbivores indicating the most effective macroalgal control and subsequent restoration of degraded coral reefs may depend on the recovery of both herbivorous fishes and *D. antillarum*.

## INTRODUCTION

Coral reefs are in decline globally with losses in coral cover and reductions in resilience in both the Caribbean Sea and Indo-Pacific (*Hughes et al., 2003*; *Jackson et al., 2014*; *Steneck et al., 2019*; *Donovan et al., 2021*). Worldwide, the loss of herbivores and the subsequent loss of their various ecosystem functions has been implicated as one of the greatest threats to coral

reefs (*Holbrook et al., 2016*; *Shantz, Ladd & Burkepile, 2020*). Specifically, the Caribbean has experienced losses in coral cover ranging from 50% to 80% since the 1970s, and these losses are only expected to increase as stressors persist or increase (*Gardner et al., 2003*; *Jackson et al., 2014*; *De Bakker et al., 2017*; *Cramer et al., 2020*). Continued mortality of Caribbean corals in combination with increased loads of nutrients and sediments, reductions and disruption in herbivore communities, and changes in competition between corals and macroalgae have contributed to increases in dominance of macroalgae on these coral reefs (*Moberg & Folke, 1999*; *Jackson et al., 2014*; *Shantz, Ladd & Burkepile, 2020*; *Cramer et al., 2020*).

The loss of herbivores in the Caribbean, both herbivorous fishes from overfishing and the sea urchin *Diadema antillarum* due to a mass mortality event in 1983, has been particularly devastating (*Lessios, Robertson & Cubit, 1984*; *Burkepile & Hay, 2011*; *Jackson et al., 2014*; *Adam et al., 2015*; *Lessios, 2016*). This may be due to inherently lower herbivore biomass and diversity creating a lower functional redundancy in the Caribbean as compared to the Indo-Pacific (*Roff & Mumby, 2012*; *Bonaldo, Hoey & Bellwood, 2014*; *Mouillot et al., 2014*; *Lefcheck et al., 2019*). It is this lack of functional redundancy that has been implicated as one of the primary causes for the lower resilience of Caribbean coral reefs (*Roff & Mumby, 2012*; *Bonaldo, Hoey & Bellwood, 2014*). Models suggest that before the mass die-off of *D. antillarum*, Caribbean coral reefs were resilient to disturbance events and did not shift to an algal-dominated state, but since the 1980s this resilience has diminished markedly (*Mumby, Hastings & Edwards, 2007*).

The primary consequence of losing both herbivorous fishes and *D. antillarum* has been an increase in macroalgal cover (*Carpenter, 1988*; *Edwards et al., 2014*; *De Bakker et al., 2017*). Increases in macroalgae have been linked to a variety of detrimental effects on coral reefs, both direct and indirect. Direct effects include increased mortality and decreased growth, fecundity, and recruitment of corals (*McCook, Jompa & Diaz-Pulido, 2001*; *Box & Mumby, 2007*; *Rasher & Hay, 2010*; *Fong & Paul, 2011*; *Donovan et al., 2021*). Indirect effects include the loss of suitable substrate for recruitment of corals and other sessile organisms, further impeding recovery of coral reefs (*Kuffner et al., 2006*; *Ritson-Williams et al., 2009*; *Precht et al., 2020*). Increased macroalgal cover and biomass in combination with declines in coral cover have persisted in modern surveys in most locations (*Jackson et al., 2014*; *De Bakker et al., 2017*).

Although many coral reefs remain in decline, there have been some positive changes in herbivore populations. Some areas have seen increases in *D. antillarum*, and these increases have been linked to lower macroalgal cover (ranging from 5-10 times lower) and increases in coral cover and recruitment (2-10 times higher) (*Edmunds & Carpenter, 2001*; *Idjadi, Haring & Precht, 2010*; *Kramer et al., 2015*; *Precht et al., 2020*). Along with increases in *D. antillarum*, increased prevalence of marine protected areas and bans on fishing herbivorous fishes have been linked to increases in both biomass and density of herbivorous fish, which in some areas has led to decreases in macroalgae (*Halpern, 2003*; *Jackson et al., 2014*; *Edwards et al., 2014*; *McField et al., 2020*; *Mumby et al., 2021*). The effect of simultaneous *D. antillarum* and herbivorous fish recovery is not well understood.

Herbivorous fishes and *D. antillarum* appear to create different macroalgal communities, likely due to their differing feeding preferences (*Adam et al., 2015*; *Adam et al., 2018*). Herbivore-macroalgal interactions are dependent, in part, on defenses employed by the macroalgae. Some macroalgae employ defenses including structural defenses, such as calcification, leathery consistency, and general toughness (*i.e., Turbinaria* spp., *Sargassum* spp.) as well as secondary metabolites that act as chemical defenses that deter herbivores (*i.e., Dictyota* spp., *Laurencia* spp., cyanobacteria) (*Littler, Taylor & Littler, 1983*; *Fong & Paul, 2011*). Many species employ both types of defenses that can act synergistically (*i.e., Halimeda* spp.) (*Hay, Kappel & Fenical, 1994*). When present, *D. antillarum* individuals act as the generalists of the system by eating most of the algae they encounter and eating many macroalgae that fishes find unpalatable (*Littler, Taylor & Littler, 1983*; *Morrison, 1988*; *Adam et al., 2015*; *Spiers et al., 2021*; *Spiers & Frazer, 2023*). While *D. antillarum* are more generalist consumers than herbivorous fishes they do show preferences to avoid macroalgae with structural defenses and some chemically rich species (*Littler, Taylor & Littler, 1983*; *Campbell et al., 2014*; *Spiers et al., 2021*). Among herbivorous fishes, *Sparisoma* parrotfish are known to target macroalgae, especially green and brown macroalgae, *Scarus* parrotfish suppress turf algae and other benthic filamentous algae, and the diet of surgeonfish, *Acanthurus* spp., is more extensive including detritus, turf and macroalgae, especially red algal species (*Dromard et al., 2015*; *Adam et al., 2015*; *Duran et al., 2019*; *Burkepile et al., 2022*). This variability in feeding preferences of herbivorous fishes accounts for some of the variability seen in feeding rates and preferences in different habitats (*Lewis, 1985*; *Ritter et al., 2021*).

An understanding of macroalgal makeup as it influences susceptibility to consumption and subsequent macroalgal community structure is particularly important due to the differing capacities of macroalgae to affect sessile organisms, particularly corals. Many macroalgal species, particularly *Dictyota*, *Lobophora*, and some cyanobacteria species, have been shown to be detrimental to the settlement and survival of coral larvae while other macroalgae have no discernable effect on coral settlement (*Kuffner et al., 2006*; *Chadwick & Morrow, 2011*; *Bonaldo & Hay, 2014*; *Ritson-Williams, Arnold & Paul, 2020*). Others, such as crustose coralline algae, can act as preferred settlement substrates (*Kuffner et al., 2006*; *Ritson-Williams, Arnold & Paul, 2016*; *Ritson-Williams, Arnold & Paul, 2020*). These differences in interactions in coral larvae can be extended to interactions between established corals and macroalgae, with strength of interaction varying among species (*Rasher & Hay, 2010*; *Fong & Paul, 2011*; *Ritson-Williams, Arnold & Paul, 2020*). Because of these differences, it is vital to understand how the composition of macroalgal communities is related to the variety and abundance of herbivores in the ecosystem. A better understanding of feeding preferences allows for better predictions of algal communities and reef resilience under different herbivore regimes.

In this study, a series of experimental feeding assays were carried out to answer questions related to feeding preferences of herbivorous fishes and the sea urchin *D. antillarum* on common Caribbean macroalgae and seagrass. Although recent studies have examined feeding preferences of *D. antillarum* or herbivorous fishes, studies that compare these major herbivore groups have not been conducted since before the *D. antillarum* die-off

**Spiers and Frazer (2023)**, *PeerJ*, DOI 10.7717/peerj.16264    3/28

of the 1980s. We hypothesized that these herbivores would show differing macroalgal feeding preferences, leading to complementarity between these herbivore types. Questions addressed with these experiments include: (1) How do feeding preferences differ between herbivorous fishes and *D. antillarum*? (2) How do feeding preferences differ among herbivorous fish species? (3) What are the broader ecological implications for differences in dietary preferences between herbivore groups?

## MATERIALS & METHODS

To better understand dietary preferences of different herbivores, a series of feeding trials were conducted with nine macroalgal species and one species of seagrass (collectively referred to as macrophytes) collected around Little Cayman, Cayman Islands (Fig. 1). Feeding trials with herbivorous fishes were conducted near Grape Tree Bay at a shallow site on the back reef behind the reef crest near a natural channel (1–2 m) (19°41′41.8″N 80°03′52.0″W) and at a deep site on the forereef (16–18 m) (19°42′01.3″N 80°03′39.6″W) (Fig. 2). Two sites were selected to compare different fish assemblages; both sites were surveyed and had numerous herbivorous fishes. *D. antillarum* feeding trials were conducted in laboratory aquaria at the Central Caribbean Marine Institute (CCMI). Work was conducted under permit from the Cayman Islands Department of the Environment (permit numbers PSAP010 and PSAP017).

Macrophytes were chosen based on their known occurrence on coral reefs in the waters surrounding Little Cayman and many other Caribbean locations. Initial surveys were conducted at the experimental sites as well as other sites surrounding Little Cayman by swimming a ~20 min roving survey and noting all conspicuous macroalgal species. Surveys of the shallow reef found *Galaxaura* spp., *Dictyota* spp. and *Halimeda* spp. to be the most common macroalgae with *Laurencia* spp., *Lobophora* spp. (decumbent and ruffled), *Padina* spp. (seasonally), *Turbinaria* spp., *Stypopodium zonale* (seasonally), and various cyanobacteria also present. The deep site had primarily *Dictyota* spp., *Stypopodium zonale* (seasonally)*,* and decumbent form *Lobophora* spp. These findings are consistent with other surveys in the Cayman Islands that found *Dictyota* spp. and *Lobophora* spp. to make up ~38% of the forereef benthos (*Dell et al., 2020*).

### Macrophyte collection

Macroalgae were photographed, identified based on morphology, and vouchered (Fig. 1). To examine the role of defenses on feeding preferences of herbivorous fishes and *D. antillarum*, these species were categorized as palatable, chemically defended, structurally defended, or both chemically and structurally defended based on literature reports (Table 1). The chemically defended algae consisted of *Dictyota* spp. and *Laurencia* sp. 2 while *Turbinaria* sp. was the only structurally defended alga and *Galaxaura* sp. and *Halimeda tuna* were both chemically and structurally defended (*Littler, Taylor & Littler, 1983*; *Paul & Hay, 1986*; *Fong & Paul, 2011*). Although some species can be chemically defended, *Lobophora* sp. was predicted to be palatable due to its ruffled morphological form and collection location in a seagrass bed (*Coen & Tanner, 1989*). *Padina* sp., *Palisada* sp., and *Thalassia testudinum* were considered palatable based on previous feeding assays (*Hay,*

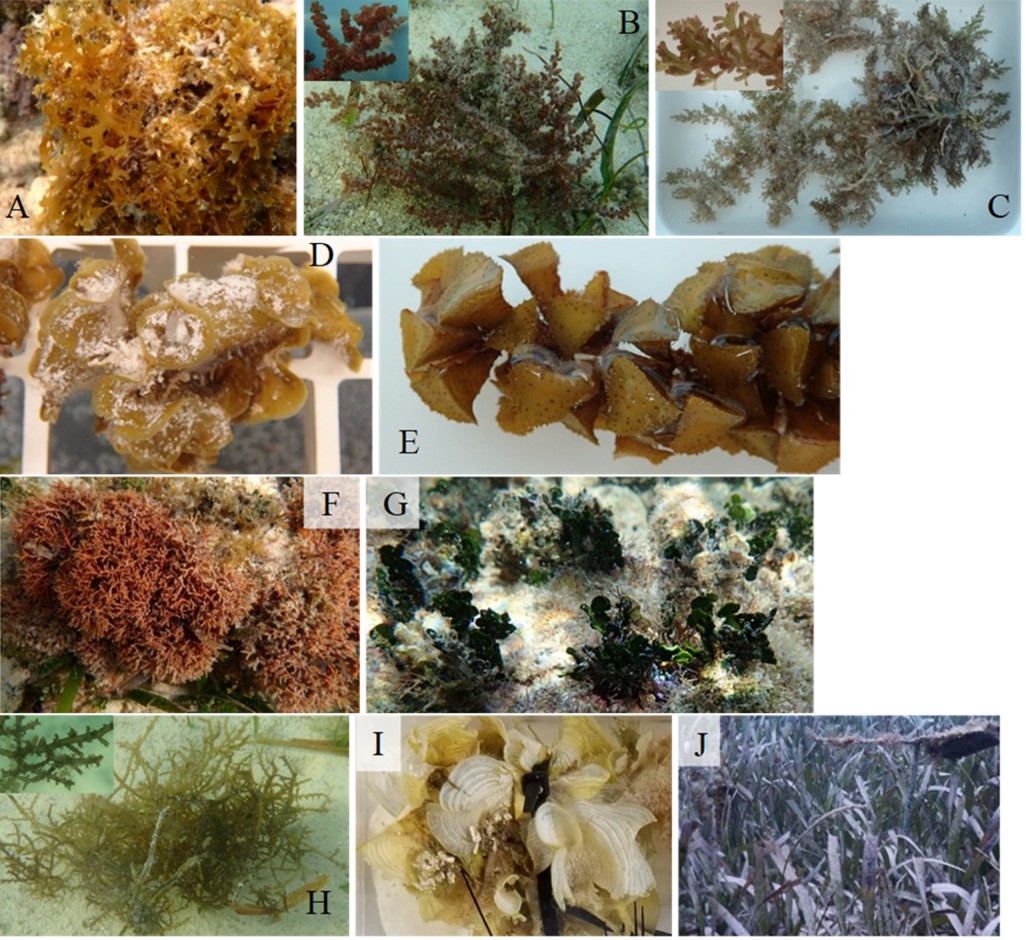

**Figure 1** **Photographs of macrophytes used in feeding assays.** Photographs show *Dictyota* sp. (A) *Palisada sp.* (B) *Laurencia* sp. 2 (C) *Lobophora* sp. (D) *Turbinaria* sp. (E) *Galaxaura* sp. (F) *Halimeda tuna* (G) *Laurencia* sp. 1 (H) *Padina* sp. (I) and the seagrass *Thalassia testudinum* (J) Insets are included to show details of some of the macroalgae.

*1984*; *Lewis, 1985*; *Paul & Hay, 1986*; *Burkepile et al., 2022*). *Laurencia* sp. 1 was unknown at the time of collection but suspected to be palatable due to the collection location in an area between the seagrass bed and the back reef. We did not collect macrophytes that were highly epiphytized but some epiphytes were present. Macrophytes were offered to the herbivores in the same state they were collected instead of being cleaned of epiphytes to avoid damage to the macrophytes and allow for the most accurate representation of how these herbivores would respond when naturally encountering the macrophytes in the field.

All algae were identified to the lowest taxonomic level; however, a high level of cryptic speciation reported for some taxa resulted in a tentative taxonomic designation. It is possible, for example, the two "*Laurencia* spp." may be other genera within the *Laurencia* species complex which has expanded due to taxonomic work to include genera other than *Laurencia* (*Rousseau et al., 2017*). *Lobophora* sp. (ruffled form characteristic of *L. variegata*)

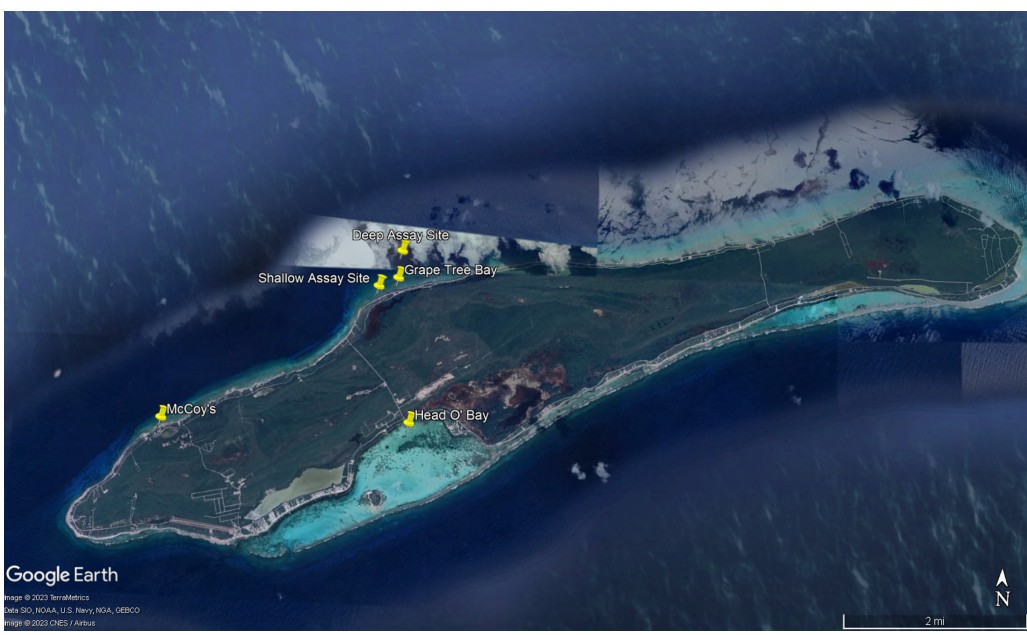

**Figure 2** **Map of study sites referenced in manuscript.** Map of experimental sites around Little Cayman, Cayman Islands including feeding assays sites, algal collection sites, and *Diadema antillarum* collection sites. Map data ©2023 Google.

(*Vieira et al., 2020*) and *Laurencia* sp. 1 were collected on the south side of the island in Head O'Bay at a depth of 1 to 2 m in an area that included seagrass beds, patchy rubble, and coral back reef habitat (19°40′29.3″N 80° 03′31.4″W) (Fig. 2). *Laurencia* sp. 1 was chosen because, during collection, it was thought to be *Acanthophora spicifera*, a known palatable alga, but on closer examination it lacked some of the distinguishing morphological characteristics of that alga. After consultation with phycologists this alga was determined to likely be part of the *Laurencia* species complex although a species identification could not be made. The seagrass *T. testudinum* and *Palisada* sp. (likely *Palisada perforata* formerly *Laurencia papillosa)* were collected in seagrass beds adjacent to the back reef at a depth of 1–3 m in Grape Tree Bay (19°41′46.6″N 80° 03′41.1″W) (Fig. 2). *T. testudinum* acted as a positive control in feeding assays as it is a known palatable macrophyte and is routinely used in similar feeding assays that have been used to examine grazing intensity on tropical reefs (*Hay, 1984*). Furthermore, *T. testudinum* is commonly found in areas surrounding shallow reefs and can be important in the diets of *D. antillarum* and herbivorous fishes (*Ogden, Brown & Salesky, 1973*; *DiFiore, 2017*). The macroalgae *Dictyota* sp. (likely *D. bartayresiana*), *Galaxaura* sp. (likely *G. rugosa*), *Halimeda tuna*, *Laurencia* sp. 2 (likely *L. microcladia*), *Padina* sp., and *Turbinaria* sp. were collected in back reef habitat at a depth of 1–3 m in Grape Tree Bay in an area where herbivorous fishes and *D. antillarum* were present. The macroalgae collected in the seagrass beds were predicted to be more palatable than those on the reef, allowing for a range from highly preferred to unpreferred algae to be compared among herbivores (*Lewis, 1985*; *Bolser & Hay, 1996*). They were collected adjacent to the back reef habitat so that light, temperature, depth, and other environmental

**Table 1** **List of macrophytes used in feeding assays in the Caribbean along with their known defenses and known responses of herbivores to congeners.** Reaction of herbivorous fishes and *D. antillarum* are presented seperately. References to original papers are also included.

| Macrophyte | Defense | Reaction of Herbivorous Fishes | Reaction of *D. antillarum* | Source |
|---|---|---|---|---|
| *Dictyota* sp. | Chemical | Variable | Variable | *Littler, Taylor & Littler (1983), Paul & Hay (1986), Hay, Fenical & Gustafson (1987), Spiers et al. (2021), Ritter et al. (2021)* and *Burkepile et al. (2022)* |
| *Palisada* sp. | Palatable | Consumption | Consumption | *Littler, Taylor & Littler (1983), Lewis (1985), Paul & Hay (1986)* and *Burkepile et al. (2022)* |
| *Laurencia* sp. 1 | Palatable | Consumption | Variable | *Littler, Taylor & Littler (1983)* and *Burkepile et al. (2022)* |
| *Laurencia* sp. 2 | Chemical | Variable | Variable | *Littler, Taylor & Littler (1983), Lewis (1985), Paul & Hay (1986)* and *Burkepile et al. (2022)* |
| *Lobophora* sp. | Palatable (ruffled form) | Consumption (ruffled)Variable (no form specified) | Consumption (no form specified) | *Paul & Hay (1986),Morrison (1988), Coen & Tanner (1989)* and *Burkepile et al. (2022)* |
| *Turbinaria* sp. | Structural | Consumption | Avoidance | *Littler, Taylor & Littler (1983), Lewis (1985)* and *Spiers et al. (2021)* |
| *Galaxaura* sp. | Chemical & Structural | Variable | Not tested | *Paul & Hay (1986)* |
| *Halimeda tuna* | Chemical & Structural | Avoidance | Avoidance | *Littler, Taylor & Littler (1983), Paul & Fenical (1986), Paul & Hay (1986)* and *Ritter et al. (2021)* |
| *Thalassia testudinum* | Palatable | Consumption | Consumption | *Hay (1984), Lewis (1985), Paul & Hay (1986)* and *Ritter et al. (2021)* |
| *Padina* sp. | Palatable | Variable | Not Tested | *Lewis (1985), Paul & Hay (1986), Coen & Tanner (1989)* and *Burkepile et al. (2022)* |

factors were similar among the shallow reef and seagrass habitats. All these macrophytes were selected due to their high abundance in the area as well as variation in defense mechanisms.

## Fish feeding assays

Fish feeding assays were conducted at the shallow and deep sites in September and October 2017. All assays ran for 24-hours after which they were scored and collected. Shallow feeding assays were initially set up between 1400 hrs and 1600 hrs along a 35-m-long area representing the transition zone between the reef edge and sandy bottom. This location allowed the feeding assays to remain underwater throughout the tidal cycle. Feeding assays at the deep site were deployed between 1000 hrs and 1200 hrs by placing them at the top of reef spurs so that macrophytes were easily seen and accessible to reef fishes.

Feeding assays followed methods developed by *Hay (1984)* that have subsequently been used to conduct fish feeding studies on coral reefs worldwide (*Paul & Hay, 1986*; *Steinberg & Paul, 1989*; *Ritter et al., 2021*). The taxa used for the fish feeding assays were *Dictyota* sp., *Laurencia* sp. 1, *Laurencia* sp. 2, *Lobophora* sp., *Palisada* sp., *Turbinaria* sp., *Galaxaura* sp.,

*H. tuna*, and the seagrass *T. testudinum*. Efforts were made to use the common species of macroalgae from the shallow reef site, most of which were also present on the deep reef. Thus, species offered should have been available to the fishes for background consumption and these were not novel food items. Despite *Padina* sp. being collected for an additional *Diadema antillarum* feeding assay at a later date, this species was not used for any of the original fish or urchin feeding assays as it was not present on the reef at the time of these experiments.

Fish feeding assays were conducted by attaching pieces of algae or seagrass to 80-cm pieces of braided, 3-strand, light blue polypropylene line and then attaching lines to the substrate by twisting open a portion of the line and placing it over a piece of dead coral or heavy rubble (Figs. S1A and S1B). Lines were placed at least 2 m apart along the experimental reef. All pieces of macrophyte were of the same length, approximately four cm, but did not have the same biomass since pieces were scored as eaten or uneaten and not weighed. Macrophyte pieces of the same length and similar size were used to be consistent in visual presentation and accessible to fishes as they approached the lines. Four different types of macrophytes were placed on each line in each trial except for one trial where it was necessary to place five pieces on the lines to ensure all species were used in at least three trials. The pieces on the line were suspended vertically in the water column accessible to herbivorous fishes (Figs. S1A and S1B). Pieces were evenly but randomly spaced ∼8–10 cm apart along the top two thirds of the line. Each type of algae and the seagrass were used in 3–4 feeding trials at each depth in different combinations. This was done because palatability can vary between assays when different combinations of species are offered, and multiple assays can be used to assess the effect different combinations may have on the amount eaten of each macrophyte species (*Paul & Hay, 1986*). Combinations of macrophytes for each trial were created haphazardly based on species availability and ensuring each combination was a unique one. Although we were not able to combine every species with every other species, using each species multiple times in different assays achieved our goal of gaining a more comprehensive understanding of the palatability of these macrophytes in different treatment combinations. A total of seven feeding trials were conducted at each depth and 15 lines were used in each feeding trial. Feeding trial refers to each separate deployment of 15 feeding lines. Each line within each trial had the same combination of macrophyte species although species were in different positions along the line to minimize effect of position on feeding choice.

After 24 h, the macrophyte pieces on each line were examined and scored as either eaten (completely gone) or uneaten, and all lines were collected. For this experiment, pieces were scored as eaten if no biomass outside of the line remained and uneaten if biomass remained. There was no incidence of partially eaten pieces; pieces were either consumed entirely or showed no signs of consumption. Dependent on weather conditions, new trials were either set up immediately following the conclusion of previous trials on the same day or up to 5 days later. To analyze differences in consumption of the different macrophytes after 24 h within each trial, number of pieces of each macrophyte eaten or uneaten for each feeding trial was analyzed in R version 4.0.2 (*R Core Team, 2020*) using a G-test (*Hervé, 2019*). Fisher's exact test was used as a post-hoc analysis to analyze differences

in consumption between macrophytes within each trial (*R Core Team, 2020*) following methods used in previous studies (*Hay, 1984*; *Paul & Hay, 1986*). A Bonferroni correction was used to account for multiple comparisons between macrophytes for each trial (*p*-value <0.008, <0.005 for assay G). These methods were used to allow for comparisons with results from previous studies.

In addition to the G-test used to compare consumption within trials, generalized linear mixed models in package glmmTMB in R (*Brooks et al., 2017*) were used to assess the effect of the fixed factors, *i.e.,* site and algae, on the proportion eaten while also accounting for the random factor of trial (glmmTMB(Eaten ~Algae*Site + (1|Trial), family = binomial), logit scale). Individual ropes served as replicates and were not tracked when scoring assays and therefore could not be tested as a random factor. *Halimeda tuna* and *Galaxaura* sp. were not used for these models due to being completely uneaten at all sites and uneaten at the shallow site, respectively. The formula simulateResiduals in package DHARMa (*Hartig, 2021*) was used to check goodness of fit for the model. Estimated marginal means from package emmeans (*Lenth, 2021*) were used to compare differences in the proportion eaten of each macrophyte within each site and compare the proportion eaten of each macrophyte between the shallow and deep sites.

Surveys were done at each site to determine the natural fish assemblage by laying out a transect line along the portion of the reef where the fish feeding assays were conducted. All fish in the open were identified and counted within 1 m of each side of the transect line and in the water column that extended approximately 3 m from the bottom at the deep site or to the surface at the shallow site. At the shallow site, surveys were conducted on one day along a 42-m section of the reef. Marked increases in the abundance of parrotfish, chubs, and surgeonfish were observed at this site beginning in the late afternoon (L Spiers, pers. obs., 2017) therefore replicate surveys were completed at 1030 hrs ($n = 2$) (42 × 2 m belt) and 1600 hrs ($n = 2$) (42 × 2 m belt) to capture this temporal variation in the fish assemblage over the reef during the day. At the deep site because there was no observed temporal variation in herbivore community, three 30 × 2 m belt transects were conducted on one afternoon at 1400 hrs.

In October 2018, short feeding trials were run for the purpose of recording herbivory and identifying which fish species took bites from each macrophytes species. Video recording was not possible during the initial feeding assays because video cameras were not available. Approximately 1 h of video was recorded using GoPro® cameras set up at each site next to a set of three lines. Two cameras were used for each feeding trial so that a total of six lines were observed. Lines were set up in the same way as in initial trials with 4 species of macrophytes per line in each trial with each macrophyte species used in at least three trials for a total of seven trials at each depth. These recordings were used to document the consumption of algae and seagrass by different fishes and to quantify the number of bites (standardized to bites h$^{-1}$). Bites were presented graphically by fish species as percentage of bites each herbivorous fish species took on each macrophyte species rather than number of bites because different herbivorous fish species take different sized bites, and for each fish species to consume the piece of macrophyte vastly different numbers of bites were

taken. This information was used to examine herbivorous fish feeding preference for each macrophyte species.

### Diadema antillarum choice feeding assays

A series of feeding assays were conducted with *D. antillarum* to characterize the dietary preferences of this sea urchin especially in comparison to herbivorous fishes. Specifically, we conducted ex-situ choice feeding assays that presented pieces from four different macrophyte species simultaneously (Fig. S1C). Most feeding assay trials were conducted in December 2017, with a fewer number of trials conducted in May 2018. All *D. antillarum* for feeding assays were collected from a shallow rocky site known as McCoy's (19°40′27.8″N 80°05′50.9″W) (Fig. 2). The mean test diameter ± standard error of collected *D. antillarum* was 6.5 ± 1.5 cm. At least 30 sea urchins were maintained in a flow-through seawater tank at the outdoor laboratory at the Central Caribbean Marine Institute (CCMI). This area was covered by shade cloth intended to create a light regime similar to that found underwater. While in the holding tank, urchins were fed a mix of known palatable algae to keep them well fed because starved urchins are less selective (*Cronin & Hay, 1996*). Keeping urchins fed also facilitated a better comparison to herbivorous fishes that were presumably also well fed on the reef and therefore selective. It was only after at least 24 h in the holding tank that sea urchins were used in feeding assays. Assays were done in aquaria to ensure that all macrophytes were available for consumption by *D. antillarum*, a guarantee not possible with in-situ feeding assays due to sparseness of sea urchins at many locations, which was not a problem found with herbivorous fish communities on our chosen reef sites.

For all *D. antillarum* choice feeding assay trials, pieces of four different macrophyte species of equal wet weight (4 g ± 0.4 g) were placed in each container with an urchin (Fig. S1C). The four different taxa were haphazardly placed in the corners of the tank before the urchin was added to the middle of the tank to limit the influence of distance on choice. Pieces of the same four taxa also were placed in control tanks with no sea urchins present. To set up each trial, each urchin was collected from the holding tank, its test diameter was measured to the nearest half centimeter using calipers, and then it was placed in a newly cleaned, flow-through 4-liter plastic aquarium. Fifteen urchins were used for each trial, with no urchin used on two consecutive days. All macrophyte pieces were weighed at the beginning and end of the experiment by spinning individual pieces 15 times in a salad spinner, measuring the weight of these pieces, and then immediately placing pieces in seawater to prevent desiccation.

Choice feeding assays were conducted over two time periods. In December 2017, the same macrophytes as those used in the fish feeding assays were tested: *Dictyota* sp., *Palisada* sp., *Laurencia* sp. 1, *Laurencia* sp. 2, *Lobophora* sp., *Turbinaria* sp., *Galaxaura* sp., *H. tuna,* and *T. testudinum* and equal amounts of each species were used for each trial. These macrophytes were common in shallow areas surrounding the island and were likely previously encountered by *D. antillarum* and therefore not novel food options. A total of seven different *D. antillarum* feeding assays were conducted in December to ensure that all experimental macrophytes were used at least three times and allow for a variety of combinations of macrophytes. The combinations for each trial were determined using

a random number generator. A total of 15 sea urchins were used in each trial with one urchin per aquarium. Because of restrictions on the number of *D. antillarum* permitted for this study, urchins were reused in different trials in December. In each case, after being used in a feeding trial, urchins were returned to a 100-gallon holding tank, where they fed on macroalgae for at least 24 hrs. Results of the seven trials are reported in the order they were conducted to examine if changes occurred in preference of macrophytes over time or between trials. Similar methods have been used in other studies of *D. antillarum* feeding preferences and indicated that carry-over effects were not observed between assays (*Hay, Fenical & Gustafson, 1987*).

An additional *Diadema* choice feeding assay was conducted in May 2018 to directly address the differences in feeding preference on species with known defenses. This assay used one palatable alga, *Padina* sp., one chemically rich alga, *Dictyota* sp., one structurally defended alga, *Turbinaria* sp., and one alga that employs both chemical and structural defenses, *H. tuna*. Four trials were run with ten urchins per trial and each urchin in individual aquaria. Each urchin received 20 g ($\pm$ 0.6 g) of each alga to ensure no alga was completely consumed during the 24 h assay (Fig. S1C). These trials used 40 individual urchins collected from McCoy's (10 for each trial), and no sea urchin was used more than once. Because urchins were not reused, all May trials were combined for analysis.

After 24 h, trials were considered complete, each urchin was removed, and any remaining macrophytes were collected and reweighed. After use in a feeding trial, urchins were exchanged for urchins that had been feeding in the large holding tank. To ensure accuracy of results, any urchin that did not consume >5% or <98% of the offered algae was removed from further analysis as these values could represent an urchin that did not eat at all and showed no preference or would have eaten more than was provided and may have resorted to eating a less preferred species once again representing an inaccurate preference (*Erickson et al., 2006*). To determine the amount consumed and correct for possible natural changes, we used the equation: $[T_i*(C_f/C_i)]-T_f$ where $T_i$ is the initial algal mass, $T_f$ is the final algal mass, $C_i$ is the initial no-herbivore control mass, and $C_f$ is the final control mass (*Cronin & Hay, 1996*; *Erickson et al., 2006*). All choice feeding trials were compared with Friedman's Test (*Lockwood, 1998*; *R Core Team, 2020*) with Student-Newman-Keuls tests (*Ferreira, Cavalcanti & Nogueira, 2021*) employed for follow-up pairwise comparisons. Friedman's tests are standardly used to analyze these types of feeding assays that lack independence between food choice within each feeding trial and do not meet assumptions of ANOVA (*Lockwood, 1998*; *Erickson et al., 2006*; *Capper et al., 2016*). A Pearson's Correlation test (*R Core Team, 2020*) was used for each trial to examine the relationship between *D. antillarum* test diameter and the total amount of macrophytes consumed.

## RESULTS

### Fish feeding assays

When offered the nine different species of macrophytes, herbivorous fishes showed a clear inclination to consume select species over others in each feeding trial (G-tests followed by Fisher's exact tests, *p*-value < 0.008 or *p*-value < 0.005 for assay G) (Fig. 3). Generalized

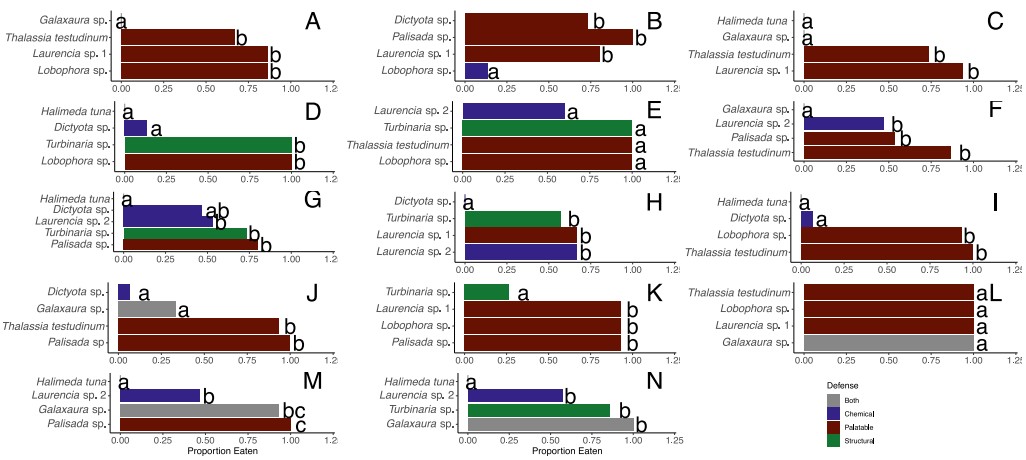

**Figure 3** **Proportion of pieces of each macrophyte eaten by fish at the shallow (A–G) and deep (H–N) sites.** Each panel represents an individual trial (*n* = 14) deployed for 24 h. Data analyzed using a G-test followed by Fisher's exact tests, and letters indicate significant differences.

mixed models revealed significant differences in proportion of each macrophyte consumed between sites and among macrophyte species (Fig. 4, Table S1). At the shallow site, *Lobophora* sp., *Laurencia* sp. 1, *Turbinaria* sp., and *T. testudinum* were high preference macrophytes while *Palisada* sp. and *Laurencia* sp. 2 were medium preference species (Figs. 3A–3G, Table 2, Table S1). In comparison, *Galaxaura* sp. and *H. tuna* were uneaten, and presumably avoided, in every trial, and *Dictyota* sp. was unpreferred with a variable amount eaten (Table 2, Table S1). Post-hoc tests of analysis of fixed factors in the generalized linear mixed effects models (emmeans()) found significant differences in proportion eaten of macrophyte species at the shallow site (Fig. 4A, Table S2). *Dictyota* sp. and *Laurencia* sp. 2 were eaten significantly less than *Lobophora* sp., *Turbinaria* sp., *Laurencia* sp. 1, and *T. testudinum* (*p*-value < 0.05) (Fig. 4A, Table S2). In addition, *Dictyota* sp. was eaten significantly less than *Palisada* sp., and *Palisada* sp. was eaten significantly less than *Lobophora* sp. (Fig. 4A, Table S2).

Surveys of fishes at the shallow site found *Acanthurus* spp. and parrotfishes to be the most abundant fishes near the benthos (Fig. 5). *Acanthurus* spp. represented 46.8% of all fishes observed, with a further 49.5% comprised of different species of parrotfish. *Sparisoma rubripinne* (initial and terminal phase) accounted for 21.6% of fishes, *Sparisoma viride* (primarily initial phase) accounted for 9.9%, and *Sparisoma aurofrenatum* (initial phase) accounted for 5.4%. *Scarus iseri* (initial and terminal phase) made up 8.1% of fishes, and *S. vetula* accounted for 3.6%.

The different herbivorous fish species were observed taking differing number of bites of the different macrophyte species (Table S3). When the data were analyzed separately for each of the herbivorous fishes that targeted macrophytes at this site, it was clear they targeted different species (Fig. 6A). *Acanthurus* spp. took almost all their bites on red algae, 47.8% of their bites on *Palisada* sp., 23.1% on *Laurencia* sp. 1, and 18.1% on *Laurencia* sp. 2. *Sparisoma rubripinne* (initial phase) took 42.6% of its bites on *Lobophora* sp., 23.3%

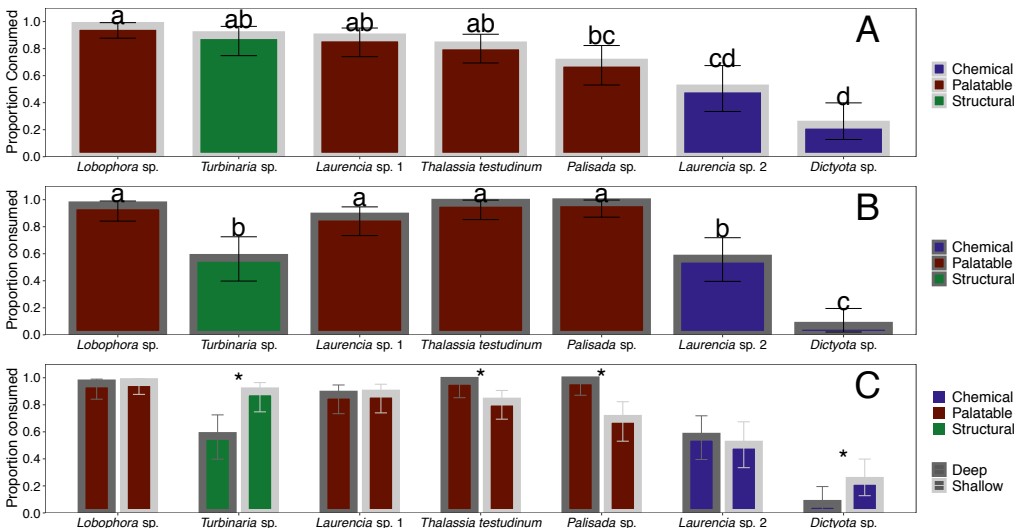

**Figure 4** **Comparison of proportion eaten among macrophyte species (*x*-axis) at the shallow (A) and deep (B) site as well as a direct comparison between sites (C).** Error bars represent 95% confidence intervals. Data were analyzed using a linear mixed effects model followed by post-hoc least-squares means tests to determine differences within and between sites. Letters indicate significant differences ($p < 0.05$) in proportion consumed within each site and * indicates significant differences for each macrophyte ($p < 0.05$) between sites.

**Table 2** **Ranking of feeding preferences of the macrophyte species for herbivorous fishes at the shallow and deep sites as well as *Diadema antillarum* preferences.** Preferences are ranked as High (75%–100% eaten), Medium (50%–75%), Low (25%–50%) or Unpreferred (0–25%). Only results from December 2017 are shown.

| Macrophyte | Herbivorous fishes-Shallow | Herbivorous fishes-Deep | *Diadema antillarum* |
|---|---|---|---|
| *Dictyota* sp. | Unpreferred | Unpreferred | Low |
| *Galaxaura* sp. | Unpreferred | High | Medium |
| *Halimeda tuna* | Unpreferred | Unpreferred | Unpreferred |
| *Laurencia* sp. 1 | High | High | High |
| *Laurencia* sp. 2 | Medium | Medium | Low |
| *Lobophora* sp. | High | High | Unpreferred |
| *Palisada* sp. | Medium | High | High |
| *Thalassia testudinum* | High | High | Low |
| *Turbinaria* sp. | High | Medium | Low |

on *Turbinaria* sp., 15.3% on *T. testudinum*, and 10.0% on *Palisada* sp. *Sparisoma viride* (initial phase) only ate two macrophytes with 93.4% of bites on *T. testudinum* and 6.6% on *Palisada* sp. The two other fishes seen biting the macrophytes were *Haemulon sciurus* and *Kyphosus* sp. which took 100% of bites from *Galaxaura* sp. and *Laurencia* sp. 1, respectively (Table S3).

At the deep site, herbivorous fish species again ate significantly more of some macrophytes than others (G-tests followed by Fisher's Exact Tests, *p*-value < 0.008)

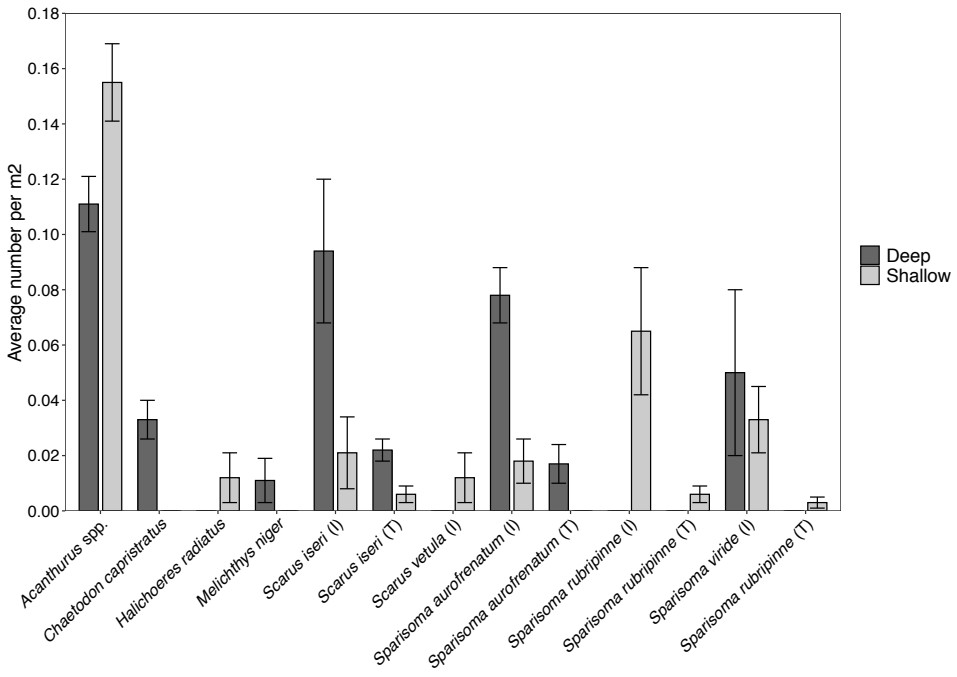

**Figure 5** **Abundance of each herbivorous fish species (±SE) per m² found at the shallow and deep site.** Values acquired from visual surveys along belt transects at the shallow (*n* = 4) and deep (*n* = 3) sites.

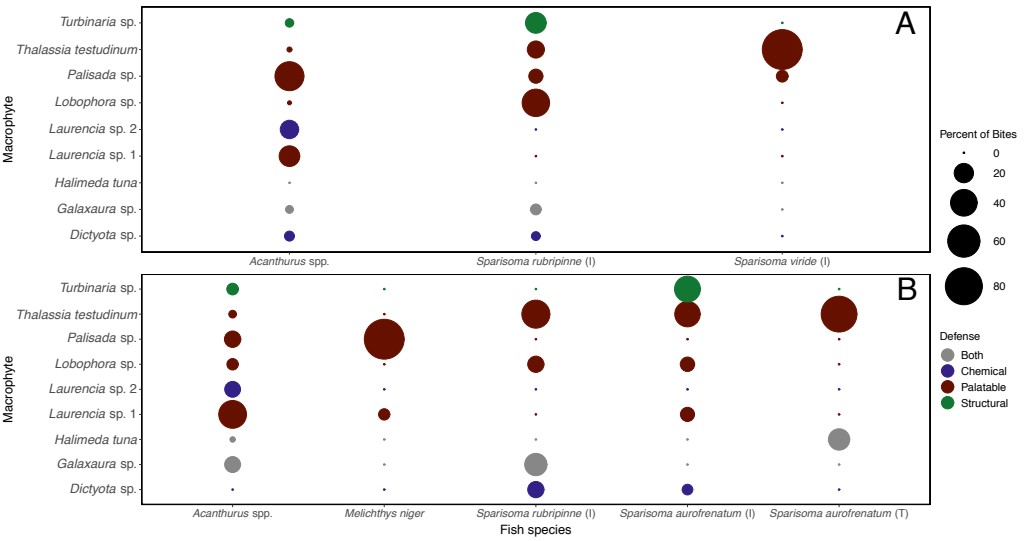

**Figure 6** **Percent of bites by each herbivorous fish on each macrophyte across both shallow (A) and deep (B) sites.** Bite numbers obtained from video recordings of the assays. For *Sparisoma* species, (I) indicates initial phase fish and (T) indicates terminal phase. Species that were recorded taking less than 1 bite per hour of macrophytes are not shown in the figure.

(Figs. 3H–3N, Table S1). Herbivorous fishes highly preferred *Palisada* sp., *Lobophora* sp., *Galaxaura* sp., *Laurencia* sp. 1, and *T. testudinum* while *Laurencia* sp. 2 and *Turbinaria* sp. were of medium preference (Tables S1, S3). Once again, *H. tuna* was not consumed, and presumably avoided, and only 0–7% of *Dictyota* sp. was consumed making it again unpreferred (Tables S1, S3). Statistical analysis of the generalized linear mixed model found *Dictyota* sp. was eaten significantly less than all other macrophytes (emmeans(), *p*-value < 0.05) (Fig. 4B, Table S2). *Laurencia* sp. 2 and *Turbinaria* sp. were eaten significantly less than *Palisada* sp., *Lobophora* sp., *Laurencia* sp. 1, and *T. testudinum* (Fig. 4B, Table S2).

Surveys conducted at the deep site found primarily *Acanthurus* spp. (26.7% of all fishes), initial phase *Scarus iseri* (22.7%), and initial phase *S. aurofrenatum* (18.7%), along with terminal phase *S. iseri* (5.3%), initial phase *S. viride* (12.0%), and terminal phase *S. aurofrenatum* (4.0%) in lower percentages (Fig. 5). The black durgon, *Melichthys niger*, was found at the site and observed eating macrophytes, but it only made up 2.7% of the fish on the transects.

When looking at the bites taken by each herbivorous fish species (Table S3), different macrophytes were targeted by different herbivorous fish species (Fig. 6B). *Acanthurus* spp. took bites of all macrophytes except *Dictyota* sp. but predominately took bites of *Laurencia* sp. 1 (43.9%). Initial phase *S. aurofrenatum* ate predominately *Turbinaria* sp. (37.7%) and *T. testudinum* (36.4% of bites). Terminal phase *S. aurofrenatum* were infrequently seen but were recorded taking bites of only *T. testudinum* (75%) and *H. tuna* (25%). Initial phase *S. rubripinne* took 44.8% of bites of *T. testudinum* and 27.6% of bites of *Galaxaura* sp. *Melichthys niger* took 93.9% of observed bites on *Palisada* sp. and 6.1% of bites on *Laurencia* sp. 1.

The types of macrophytes consumed at the deep site were similar to those at the shallow site with a few differences. Analysis of proportion eaten for each species at the shallow and deep site found statistically significant differences between sites for *Dictyota* sp., *Palisada* sp., *Turbinaria* sp. and *T. testudinum* (emmeans()) (Fig. 4C, Table S2). Specifically, for *T. testudinum* and *Palisada* sp., significantly more was eaten at the deep site as compared to the shallow site (Fig. 4C, Table S2). The opposite pattern was seen for *Turbinaria* sp. and *Dictyota* sp. where a higher proportion was eaten at the shallow site (Fig. 4C, Table S2). In addition, *Galaxaura* sp. was only eaten at the deep site, although this could not be statistically tested in this model. Proportion of pieces eaten for all other macrophytes were not significantly different between the shallow and deep site. Additionally, the abundance and diversity of herbivorous fish species seen consuming macrophytes at the deep site was greater than at the shallow site.

### *Diadema antillarum* choice feeding assay

There were significant differences between macrophytes in the amounts eaten by *D. antillarum* in December 2017 choice feeding trials (Friedman tests for each trial, *p*-value <0.05) (Figs. 7A–7G). *Laurencia* sp. 1, *Palisada* sp., and *Galaxaura* sp. were eaten significantly more than *H. tuna*, *Lobophora* sp., *Turbinaria* sp. and *T. testudinum* in all trials (Student-Newman-Keuls tests). In addition, *Laurencia* sp. 1 and *Palisada* sp. were

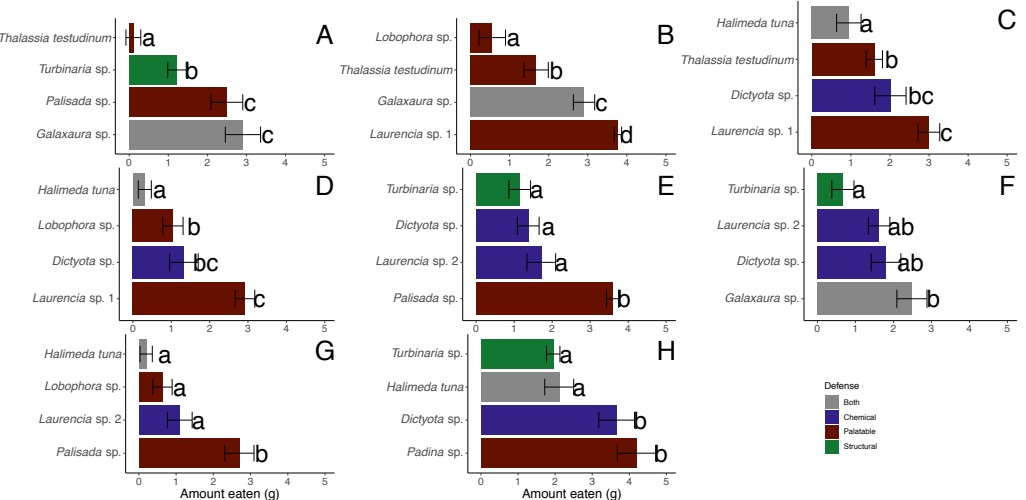

**Figure 7 Amount in grams of each macrophyte ± standard error eaten by *Diadema antillarum* in choice feeding assays in December 2017 (A–G) and May 2018 (H).** Different letters indicate significant differences when analyzed by Friedman's tests followed by Student-Newman-Keuls tests.

eaten significantly more than *Laurencia* sp. 2, and *Dictyota* sp. was eaten more than *H. tuna* (Student-Newman-Keuls tests).

When looking at the average amount consumed across all trials, the most highly consumed macrophytes were *Laurencia* sp. 1 ($n = 43$, $3.22 \pm 0.14$ g (mean ± SE)), *Palisada* sp. ($n = 39$, $2.92 \pm 0.21$ g), and *Galaxaura* sp. ($n = 43$, $2.77 \pm 0.22$ g) (Table 2). *Dictyota* sp. ($n = 56$, $1.64 \pm 0.18$ g) and *Laurencia* sp. 2 ($n = 38$, $1.50 \pm 0.19$ g) were of lower preference followed by *T. testudinum* ($n = 44$, $1.12 \pm 0.17$ g) and *Turbinaria* sp. ($n = 42$, $1.01 \pm 0.16$ g) (Table 2). *Lobophora* sp. ($n = 39$, $0.75 \pm 0.17$ g) and *H. tuna* ($n = 40$, $0.52 \pm 0.15$ g) were both unpreferred species (Table 2).

There was no significant relationship between total amount consumed and test diameters in trials 1, 3, 4, 5, 6 or 7 (Figs. 7A, 5C–5G) in December 2017 (Pearson correlation, $p$-value > 0.05). In trial 2, the amount eaten increased with greater *D. antillarum* diameter (Pearson correlation, $r$ (12) = 0.73, $p$-value = 0.0002). Specifically, larger diameter *D. antillarum* ate more of *Lobophora* sp. (Pearson correlation, $r$ (12) = 0.69, $p$-value = 0.006) and *Laurencia* sp. 1 ($r$ (12) = 0.75, $p$-value = 0.002).

When comparing the amount eaten of the different algae in the May 2018 choice assay (Fig. 7H), significantly more *Padina* sp. ($4.19 \pm 0.52$ g) and *Dictyota* sp. ($3.66 \pm 0.48$ g) were eaten compared to *Turbinaria* sp. ($1.95 \pm 0.18$ g) and *H. tuna* ($1.89 \pm 0.37$ g) (Friedman's test, $df = 3$, $p$-value = 0.04). There was no significant correlation between the total amount eaten and test diameter in these assays (Pearson correlation, $r$ (38) = 0.13, $p$-value = 0.40).

## DISCUSSION

Determination of herbivore feeding preference allows for a better understanding of how different herbivore assemblages may lead to variations in macroalgal community structure. Little Cayman offered a unique study area because 15% of the nearshore waters are in a marine protected area, which has led to abundant and diverse herbivorous fish communities compared to much of the Caribbean (*Creary et al., 2008*; *Dromard et al., 2015*). Even with this level of protection, Little Cayman lacked the large terminal phase *Scarus* parrotfish found on many reefs of the Florida Keys, which are essential for macroalgal control on Caribbean reefs (*Shantz, Ladd & Burkepile, 2020*). The relatively high herbivorous fish populations in combination with the ability to collect large adult *D. antillarum* allowed for a comprehensive view of herbivory by diverse consumers on a Caribbean island, a view that has been hard to capture for decades due to the *D. antillarum* mass die-off and overfishing. The experimental setup allowed for a comparison of feeding preferences between the two types of herbivores by tailoring each experiment to the mobility of the herbivore. Herbivorous fishes, being much more mobile than *D. antillarum*, were able to encounter our experimental macrophytes in the field much the same way *D. antillarum* encountered the macrophytes in the aquaria. In addition, the well-fed state of the sea urchins, and the capacity of herbivorous fishes to be well fed on the reef before encountering the experimental lines, allowed for each herbivore to show preferences rather than only consuming algae because they were starving.

The goal of this study was to compare the feeding choices of herbivorous fishes and the sea urchin *D. antillarum* for the purpose of determining differences and similarities in feeding preferences among the herbivores. There appeared to be substantial redundancy between the herbivorous fishes and *D. antillarum* in preference for the most palatable macrophytes, *Laurencia* sp. 1 and *Palisada* sp. *Laurencia* sp. 1 was a high preference macrophyte in all feeding assays while *Palisada* sp. was highly preferred by fish at the deep site and by *D. antillarum* and of medium preference by fishes at the shallow site (Table 2). This difference in consumption of *Palisada* sp. at the different sites may be attributed to *M. niger*, which was only found at the deep site and observed taking the majority of bites off *Palisada* sp. Both *Laurencia* sp. 1 and *Palisada* sp. were collected in a seagrass bed. Algae from low herbivory habitats, such as seagrass beds, are often highly susceptible to grazing presumably because they are less chemically or structurally defended than interspecific and intraspecific algae found outside these habitats (*Hay, 1984*; *Lewis, 1985*; *Bolser & Hay, 1996*).

There was also overlap among the macrophytes that were least consumed with *H. tuna* completely uneaten in all fish feeding assays and consumed very little in *D. antillarum* feeding assays. This result is consistent with previous studies that have found low palatability for many *Halimeda* spp., which can be rich in secondary metabolites that act as deterrents to herbivory along with their $CaCO_3$ structure that can act synergistically with chemical defenses as a deterrent (*Paul & Fenical, 1986*; *Paul & Van Alstyne, 1988*; *Hay, Kappel & Fenical, 1994*). Unlike fish, *D. antillarum* did eat a portion of *H. tuna* in all trials even when

species considered more palatable were present indicating a willingness to eat the alga. This was especially shown in the May 2018 assay, although it was not preferred.

Along with the similarities between the herbivores there were differences that could also affect the structure of macroalgal communities. The most significant difference between the herbivore groups was seen in their preference for brown algae, *Dictyota* sp., *Lobophora* sp. and *Turbinaria* sp. *Dictyota* sp. was eaten more by *D. antillarum* and herbivorous fishes consumed more *Lobophora* sp. and *Turbinaria* sp. Given the abundance of these brown algae on Caribbean coral reefs and their detrimental effects on both adult corals (*Box & Mumby, 2007*; *Rasher & Hay, 2010*; *Fong & Paul, 2011*; *Vieira, 2020*) and coral settlement (*Kuffner et al., 2006*; *Chadwick & Morrow, 2011*; *Ritson-Williams, Arnold & Paul, 2020*), understanding which herbivores consume each alga can have important implications for reef management. Species within the genus *Dictyota* are known to produce a variety of secondary metabolites, and algae pieces are often low preference for herbivorous fishes (*Bolser & Hay, 1996*; *Vallim et al., 2005*; *Fong & Paul, 2011*) but not *D. antillarum* (*Littler, Taylor & Littler, 1983*; *Solandt & Campbell, 2001*). This is consistent with our assays that showed little consumption of *Dictyota* sp. by fishes but higher consumption by *D. antillarum* in choice assays as compared to *Turbinaria* sp. and *H. tuna*. Extracts of some *Dictyota* spp. have been shown to deter feeding by *D. antillarum*, although they will eat live algae especially when offered no other food choice (*Spiers et al., 2021*). This work presented here and in other studies suggest that although *Dictyota* spp. are not preferred macroalgae, *D. antillarum* will eat these brown algae.

The high consumption of *Lobophora* sp. by herbivorous fishes but not by *D. antillarum* was unexpected due to the presence of secondary metabolites within this genus and lack of obvious structural defenses (*Targett et al., 1992*), which would predict avoidance by fishes but not urchins (*Paul & Hay, 1986*; *Coen & Tanner, 1989*). The consumption by herbivorous fishes was attributed to the collection location of *Lobophora* sp. in seagrass beds and its ruffled form (likely *L. variegata*) (*Vieira et al., 2020*), both of which have been linked to being more palatable (*Coen & Tanner, 1989*; *Bolser & Hay, 1996*). The consumption of *Lobophora* sp. by herbivorous fishes in the Cayman Islands is consistent with other studies that also found *Acanthurus* species and some *Sparisoma* species readily consume this alga (*Lewis, 1985*; *Dell et al., 2020*; *Burkepile et al., 2022*). Specifically, in our study, it appears the *S. rubripinne* may have preferentially eaten this species. It is not entirely clear why *Lobophora* sp. was less preferred by *D. antillarum*, but it was probably due to its ruffled, ball-like structure and thicker thallus (*Coen & Tanner, 1989*). Similarly, a decumbent species of *Lobophora* and its chemical extract were unpalatable to *D. antillarum* in the US Virgin Islands (USVI) (*Spiers et al., 2021*).

The other brown alga tested, *Turbinaria* sp., was categorized as structurally defended due to its thick leathery/rubbery branches and a thick-walled, heavily corticated structure (*Littler, Taylor & Littler, 1983*). Although this structure did not deter the herbivorous fishes, particularly *Sparisoma* parrotfish at the shallow site, it did deter feeding by *D. antillarum*, and this deterrence was particularly apparent in May 2018 where it was one of the least consumed species. In addition to *Sparisoma* parrotfish, *Acanthurus* spp. were also seen biting at the *Turbinaria* sp., especially at the deep site, but due to the lack of appreciable

consumption it is likely these fish were targeting epiphytes associated with the alga rather than eating the alga itself. *Turbinaria* sp. was eaten significantly more at the shallow site compared to the deep site. These differences could be related to differences in fish species composition and their feeding preferences at the two sites. Previous studies have shown *Sparisoma* parrotfish to readily consume *Turbinaria* sp. (*Littler, Taylor & Littler, 1983*; *Lewis, 1985*).

*Galaxaura* sp. was not consumed by fish at the shallow site but almost completely consumed at the deep site, which may be linked to the presence of Black Durgon (*Melichthys niger*). These fish were only seen at the deep site and were observed by divers biting *Galaxaura* sp. on the lines, although this was not caught on camera. *M. niger* is a generalist omnivore which has been observed consuming significant amounts of macroalgae in Caribbean locations (*Mendes et al., 2019*; *Tebbett et al., 2020*). Specifically, our report is not the first observation of *M. niger* consuming *Galaxaura* sp., and in previous assays *M. niger* appeared to preferentially consume *Galaxaura* sp. (∼80% of *M. niger* bites) (*Tebbett et al., 2020*). This study cited *M. niger* as the dominant herbivore, a condition not replicated in our feeding assays. Both our study and that by *Tebbett et al. (2020)* were conducted off Little Cayman Island and were geographically close to each other on the north side of the island. It is possible that Little Cayman represents an isolated enough island that *M. niger* has taken on a more herbivorous diet as has been seen on other isolated reefs (*Mendes et al., 2019*). This observation adds to the collection of studies showing that unique and/or rare species may play a unique role as herbivores on taxa of macroalgae often avoided by more traditional browsers (*Bellwood, Hughes & Hoey, 2006*; *Dell et al., 2020*; *Burkepile et al., 2022*). In our study, *Acanthurus* species also were seen feeding on *Galaxaura* sp., although they did not appear to completely consume it during the 1-hour videos like they did other algal species, and *S. rubripinne* was also observed feeding on *Galaxaura* sp., particularly at the deep site. *Galaxaura* species are known to be both chemically rich and calcified, but in these assays, this combination of defenses was not as effective in deterring herbivorous fishes as might have been expected from previous studies (*Paul & Hay, 1986*). These results are however consistent with studies showing surgeonfish consume *Galaxaura* spp. (*Duran et al., 2019*). *Galaxaura* was the third most eaten in *D. antillarum* choice assays, a relatively high consumption that is consistent with previous feeding assays (*Solandt & Campbell, 2001*).

Differences in food preference facilitated examination of functional complementarity in herbivory in Little Cayman. Although, *D. antillarum* showed the ability to eat all types of macrophytes, there were less preferred algae, such as *Turbinaria* sp. and *Lobophora* sp., that typically employed some form of structural defense. Herbivorous fishes, particularly parrotfishes, were more selective in what they consumed and avoided some algae that employ chemical defenses. In contrast, *Acanthurus* species consumed some chemically rich genera such as *Galaxaura* sp. and *Laurencia* spp., but they did not graze to the same extent on thick-bladed species such as *Turbinaria* sp. and *Thalassia testudinum*. This difference in feeding by herbivorous fishes can be attributed to differences in mouth structure. Parrotfish have large beaklike mouths that allow them to engulf and consume large portions of macrophytes, and *Sparisoma* parrotfish are known to completely tear algae away from

substrate (*Bonaldo, Hoey & Bellwood, 2014*; *Adam et al., 2018*). In comparison, *Acanthurus* species have relatively small mouths that do not open wide, which leads many species to crop algae through quick nips at their top portions (*Purcell & Bellwood, 1993*; *Dromard et al., 2015*; *Adam et al., 2018*; *Tebbett, Siqueira & Bellwood, 2022*). This difference in mouth size is consistent with the type of macrophytes that these two groups of herbivorous fishes consumed. *Acanthurus* species were seen eating almost entirely branching red algae, some of which were calcified and/or chemically rich. In comparison, *Sparisoma* parrotfish ate the brown algae *Turbinaria* sp. and *Lobophora* sp. and large bladed seagrass *T. testudinum*. These differences in feeding preferences among herbivorous fishes are consistent with previous experiments that showed dietary complementarity between *Acanthurus* spp. and *Sparisoma* parrotfish, specifically *S. aurofrenatum* and *S. rubripinne* (*Lewis, 1985*; *Burkepile & Hay, 2011*; *Adam et al., 2018*; *Burkepile et al., 2022*).

Differences in herbivore preferences seen between shallow and deep sites indicate that functional redundancy differs not only on the regional scale, but also between individual reefs within a region based on the composition of the herbivore community. Although there may be some functional redundancy between *Diadema antillarum* and herbivorous fishes in regard to the most and least preferred species, it is incomplete with some types of algae only eaten by one or the other type of herbivore and differences among families and species of herbivorous fishes. Differences in complementarity and redundancy with regard to feeding among herbivores highlights the importance of maintaining biodiversity and suggests that it will be important to have a broad suite of herbivores on coral reefs if the goal is to control algal abundance in order to facilitate recovery of coral reefs (*Lefcheck et al., 2019*).

Reefs with abundant *Sparisoma* parrotfish may be dominated by chemically rich species while reefs with *Acanthurus* species may have predominately thick-bladed or structurally defended macrophytes. *D. antillarum* dominance may lead to reefs dominated by structurally defended species such as *Turbinaria* sp. Pre die-off comparisons of algal communities in Jamaica found that fish alone resulted in the establishment and proliferation of erect algae and specifically created a community dominated by chemically rich algae (*Morrison, 1988*). Furthermore, (*Burkepile & Hay, 2008*) found that *Scarus taeniopterus* (a species not found at our sites) and *Acanthurus bahianus* allowed late successional stage macroalgae to flourish while *Sparisoma aurofrenatum* reduced the cover of upright macroalgae. In Little Cayman, both shallow and deep reefs were dominated by *Dictyota* spp. along with *Halimeda* spp. and *Galaxaura* sp. on the shallow reef, which is consistent with the primary macroalgal consumer observed being herbivorous fishes with only a very low abundance of *D. antillarum* seen at the shallow site.

Continued presence of macroalgae on many reefs, no matter the type, can impede the settlement of sessile organisms, such as corals (*Kuffner et al., 2006*; *Chadwick & Morrow, 2011*; *Ritson-Williams, Arnold & Paul, 2020*). Specifically, as it relates to the species used in these feeding assays, some *Dictyota* spp. and *Lobophora* spp. have been shown to be detrimental to coral larval settlement and survival as well as the growth of adult corals (*Kuffner et al., 2006*; *Box & Mumby, 2007*; *Fong & Paul, 2011*; *Ritson-Williams, Arnold & Paul, 2020*). In contrast, *Halimeda* spp. have also been seen to affect coral larvae but may

actually increase settlement rates, although this may vary by coral species (*Ritson-Williams et al., 2009*; *Olsen, Sneed & Paul, 2016*). These differing effects of macrophytes on corals highlights the importance of better understanding the structuring of algal communities.

## CONCLUSION

This study examines differences in feeding preferences between two major herbivore groups, *Diadema antillarum* and herbivorous fishes, as well as among herbivorous fish species, with the goal of better understanding the ecological implications of different herbivore communities. *D. antillarum* consumed some of every species of macrophyte offered but ate the most of thin red algae species and *Padina* sp. while avoiding structurally defended *Turbinaria* sp. Parrotfishes mostly ate the seagrass *T. testudinum* as well as *Turbinaria* spp., *Palisada* sp. and *Lobophora* sp. while avoiding chemically rich *Dictyota* sp. and chemically and structurally defended *H. tuna*. Surgeonfish primarily ate thin red algae, primarily *Laurencia* sp. 1, while avoiding thick bladed *T. testudinum* and *Turbinaria* sp. These results show that not all herbivores serve the same role on Caribbean coral reefs and supports the functional importance of a diversity of herbivores in maintaining algal communities. Management interventions may be necessary to ensure the presence of these and other essential herbivores on reefs. To this end, some Caribbean countries have taken steps to limit the harvest of parrotfishes while others have begun efforts to repopulate reefs with *D. antillarum* (*National Oceanic and Atmospheric Administration, 2019*; *McField et al., 2020*). This study suggests that both management practices are important simultaneously. These efforts can help reduce macroalgal abundance, which, in turn, may provide more open substrate for coral settlement, reduce stress from coral-algal interactions, and increase Caribbean coral reef resilience.

## ACKNOWLEDGEMENTS

We thank the volunteers and staff at the Central Caribbean Marine Institute for their assistance in algal collection and experimentation, particularly J. Kuehl and P. Maneval. Permits were obtained from the Cayman Islands Department of the Environment in 2017. We thank Colin Shea and Bill Sharp at the Florida Fish and Wildlife Conservation Commission for their assistance with statistical analysis and final drafts of this manuscript and the Frazer lab at University of Florida and V. Paul for comments on drafts of the manuscript.

### Funding

The authors received no funding for this work.

### Competing Interests

The authors declare there are no competing interests.
## Author Contributions

- Lindsay Spiers conceived and designed the experiments, performed the experiments, analyzed the data, prepared figures and/or tables, authored or reviewed drafts of the article, and approved the final draft.
- Thomas K. Frazer conceived and designed the experiments, authored or reviewed drafts of the article, and approved the final draft.

## Animal Ethics

The following information was supplied relating to ethical approvals (i.e., approving body and any reference numbers):

The Department of Environment of the Cayman Islands reviewed and approved all experiments conducted.

## Data Availability

The results of all experiment are available in the Supplemental Files.

## Supplemental Information

Supplemental information for this article can be found online at http://dx.doi.org/10.7717/peerj.16264#supplemental-information.

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
