# Peer review of "Comparison of feeding preferences of herbivorous fishes and the sea urchin Diadema antillarum in Little Cayman"

_PeerJ, doi:10.7717/peerj.16264_

## Round 0.1 · original submission · Major Revisions

Dear Dr. Spiers,

Three reviewers returned extensive and helpful comments, and I encourage the authors to address them thoroughly. I would encourage the authors to clarify the comparison between Caribbean and Indopacific reefs in the introduction, as well as add more details about G-tests. Finally, I would also encourage the authors to review the results section in detail since the reviewers found it very hard to follow.

All the best

Juan Pablo

Reviewer 1 ·

Basic reporting

This manuscript meets the majority of requirements, but there are some points that should be addressed prior to publication.

In multiple places information is presented in a haphazard manner that is difficult for the reader to follow. For example, considerable attention is given to a comparison of Caribbean and Indopacific reefs in the introduction (lines 44-59), yet Indopacific reefs are not part of the study and are not mentioned in the manuscript again. These differences are irrelevant to the current study so the information should instead be presented as a synthesis, rather than comparison.

Similarly, there should be some mention of the earlier work on urchin grazing since this is a central part of the study. Yet the authors simply mention this work exists and do not discuss it at all. There are also multiple places in the methods and results where information is out of place.

An important point the authors make in both the abstract (lines 21-22) and the introduction (lines 70-75) is that some reefs with recovering fish and urchin populations have experienced a decline in macroalgal cover and increase in coral cover and recruitment. Given the relevance of this to the authors’ thesis, this pattern deserves a sentence or two more explanation.

Experimental design

It seems multiple statistical analyses have been conducted on one dataset, yet the authors fail to apply the Bonferroni correction. If the authors believe the Bonferroni correction is not required, then they must justify why. It is also perplexing why there is no statistical analysis of fish versus urchin consumption when this is one of the authors’ three questions posed in the introduction.

There are also some points in the experimental design that are not adequately explained. The choice of study sites is one example. It is curious that the authors chose to establish a feeding trial on the shoreward side of the reef crest, presumably some distance from the forereef. Given that the literature they choose to include is all based on the forereef, this choice of site is perplexing and should be justified. If the fish species they record in their shallow trials are not found on the forereef, or are found in vastly different densities, then it questions whether their results are applicable to the forereef. This point does not prevent publication but should be specifically addressed in the discussion.

Similarly, the authors report that they conducted surveys to determine the most common algae on the reef, yet no methods are given. This lack of information is misleading so the authors should explain their methods more thoroughly.

The inclusion of an ‘unknown red’ alga is not informative nor an example of investigation performed to a “high technical standard”. That an unknown alga is consumed more or less than others is not meaningful and should either be identified or omitted.

The methods and results are confusingly written and need to be restructured for clarity.

Validity of the findings

Data are provided and conclusions are appropriate to the results in this study.

Additional comments

Some notes by line are included below.

Abstract
18-21 First few sentences of abstract introduce the subject well.
22 The authors should explain which reefs and this pattern in more detail.
Overall, the abstract is well-written and sets up the rest of the paper nicely.

Introduction
First few paragraphs of intro are well written and explain the situation comprehensively.

51 Small point but I’m surprised Lessios wasn’t cited here.

44 – 59 Why are you comparing the differences between the Caribbean and Indo-Pacific in the introduction when your methods and results are exclusively from the Caribbean (line 102)? This section is off topic and misleading for the reader. Either justify the relevance of the comparison or, better yet, remove it and synthesize the information, then use the space to focus on the point below.

70-75 This topic is very relevant to your thesis and gets a full sentence of the abstract but nothing more is said about it. You write that assessing the combined effects are perhaps more important than those of fishes or urchins individually, but that only applies on reefs where Diadema are recovering or exist in densities sufficient to have a measurable impact. To support your statement, you need to explain in more detail the observed pattern of coral recovery following herbivore recovery. Which studies have reported an increase in urchins and which in parrotfishes? Has the recovery of corals occurred equally following both? Which reefs have seen a recovery of both? You establish the importance of testing both urchins and fishes partially based on this pattern, so this pattern deserves more attention.

75-76 Sharp jump between topics. Advise including a linking sentence.

76-79 Would be clearer as two sentences.

93-95 Not sure what point you are trying to make here. You write, “These differences in interactions in coral larvae and juveniles”, but you haven’t mentioned juveniles at all yet, so the reader has no idea which differences you’re referring to…

102-104 What did these studies on Diadema find? This is a key point so should be explored in more detail.

Methods
114 Using seagrass should be justified here. You explain later (in line 135) but the reason would fit better the first time you introduce the species.

116 Why are these surveys not explained in the methods? Were surveys conducted or were these algal species simply the easiest to obtain? The survey methodology should be explained if surveys were indeed conducted, or the method of choosing the algae should be more accurately described.

119 You refer to the ‘deep site’ as though the reader is already familiar with your study area, yet this is the first time you’ve mentioned there’s a deep site. A site description at the beginning of the methods section would rectify this confusion.

123 Would it not make sense to refer to your photographs in Figure 1 here?

127-142 This is confusing to read and would be greatly assisted by a site description and map. Furthermore, I suggest splitting the rationale for including these species from the description of where they were collected.

143 Your statement needs support from the literature. You include references in line 448 but they need to be provided here as well.

146-147 Earlier you said you conducted surveys? This sentence would fit better near line 116. Additionally, as above, I suggest first describing the algae and the reasons you chose them. Then you can explain where you collected them while referring to a map and site description.

159-162 Good to explain this point.

165-175 What is the justification for choosing these two sites? Were herbivores in higher abundance at these sites than elsewhere? Also, 1-2m is very shallow, why not conduct the surveys on the forereef rather than on the shoreward side of the reef crest?
The literature referred to in the manuscript is focused on forereefs but the shallow site of these experiments is not on the reef. From the description it appears to be in a channel on the shore side of the reef crest, so it potentially quite far from the forereef. This isn’t a problem in itself, but it must be explained. As it is, it seems this detail is slipped in and not addressed which is very misleading for the readers. Also, how accurate are comparisons from this location with studies conducted on the forereef? The fish community could potentially be quite different in the channel from the forereef, so this needs to be addressed.

167 Why is the proximity to a coral nursery relevant?

170 & 174 The way these methods are written it sounds as though the trials are run for 2 hours. It only becomes clear later that they are run for 24. Revise for clarity.

176 This next section jumps from setting up the experiment to recording data after the conclusion of the experiment and back to the set up. Revise so the description of the methods follows chronologically and is easier for someone wishing to reproduce them.

179 Why were not all 10 algae used in the fish assays? It seems Padina is missing from this list. This omission should be explained.

186 Those photos are very helpful and would fit well as part of the main figures.

191 This sentence is out of place here. This section needs reorganized for clarity.

192 This is contradictory. Why would you need to check for visible bite marks if you scored algae as either ‘fully consumed’ or ‘unconsumed’? Furthermore, this seems a questionable methodology. Bite marks from some species, such as surgeonfishes, are very small. How can you be sure you didn’t miss these?

188-195 You jump from the experimental set-up to scoring the results of the experiment and back to the set-up again. For clarity, describe the set-up completely before moving to the scoring.

199-201 This statement should be supported by literature.

208 Good point to include in the experimental design.

209 All this information would fit better around line 170.

216 Unless I’m missing something, these same data are statistically analyzed twice so should be Bonferroni corrected.

237 Consider the following for clarity: were presented graphically by fish species as a percentage of bites taken… Otherwise it could be interpreted that you calculated the percentage out of the total bites taken of each alga.

273 It would be helpful to refer to your supplementary figure 1C here. Also, it would be good to include the average weight of algae used in this experiment. You give the value in grams for the second trial (on line 303) so why not include it here too?

289 Was random number generator used for the fish assays too? This is another example of a detail that should be included earlier. Suggest reorganizing the methods for clarity.

309 This needs to be justified. Why exclude these trials?

Results
326 It is perplexing to start the fish assay section with results from the surveys. It would assist the reader’s comprehension for your results section to follow the same order of experiments as the methods. It may be worth subdividing the feeding assays into three separate sections, one for surveys, one for your initial trials, and one for the videoed trials.

328 This is a small point but from these values it appears that Halichoeres radiatus is 3.7% of the fish community which isn’t a ‘predominant’ member of the community. Suggest revising.

333-337 Do these results apply to the shallow site exclusively? To remove any ambiguity it would be helpful to state this.

335 You give the p value here, but this should have been included in the methods and should be Bonferroni corrected.

337 Why jump to site comparison here? This result is out of place. It would make more sense to expand on sentence 333-335 first, then to discuss the deep site, and then to discuss the site comparison.

367 This sentence discusses the fish abundance at the deep site but comes between two sentences discussing fish consumption. This is another example of the need for restructuring of the manuscript.

377 To avoid ambiguity consider ‘each herbivorous fish species’ (same applies to the legend for table 3).

391 Good that you explain about Galaxaura.

Discussion
421 If this is the main point of your study then why not analyze this statistically?

459-462 Would work better as two sentences.

One paper that would be relevant to refer to is Burkepile et al. 2022 in Food webs. This paper would also be relevant to lines 85-87 in the introduction.

591 *are

Discussion is well written and does a good job of tying all the results together.

References look good.

Figures
Figure 3 would be clearer to interpret if the species in the middle panel were in the same order as the other two panels. I understand why it was arranged the way it was, but having all 3 graphs in the same configuration would be easier to interpret.

Figure 4 looks great and highlights the complementarity and redundancy among herbivores well. It would be helpful to include which experiment these data come from as you have done for table 3.

Table 1 is informative, and is a worthwhile addition to the manuscript, but could be improved because in places it is too general to be useful. Under reaction of herbivorous fishes, half of the column says ‘variable’ which is not interesting or useful and is presumably because the reaction depends on species of fish. Rather than focus at guild level, the authors could make use of the research that addresses differences among herbivorous fishes at the species level, and this column could instead include which fish species consume these algae. This research is referred to in lines 85-87 so could be incorporated in the table too.

It would also be interesting to add a column to include the results from this current study. That would provide a clear comparison of results and would show how this current work adds to the body of literature.

Table 2 These data would be better presented as a graph.

Table 3 Not sure what this table adds over figure 4. If you are wanting to limit the number of figures, then I suggest moving this to supplementary information and instead including the photos of the experiments in the main figures.

Supplementary Files
The datasets are well explained but it would be helpful to include a legend with the tables so the reader can understand them without having to first find the reference to them in the manuscript text.

Reviewer 2 ·

Basic reporting

Some references missing. See below.

Experimental design

Some limitations with analyses. See below.

Validity of the findings

No significant problems.

Additional comments

This paper is a nice investigation of the grazing preferences of fishes and Diadema. As the authors state, grazing is a key process on Caribbean reefs and while there has been a range of studies on parrotfish grazing, the preferences of Diadema are poorly documented. The summary Table 4 will be a nice addition to the literature.

I have only 1 major comment. I am not very familiar with G-tests, but I wonder whether the clustering of the data (by trial and by rope) are taken into account. The individual algal pieces are clearly not independent replicates and I am curious if that is problematic for the assumptions of G-tests (it’s also not clear how it is accounted for in the urchin trials). Furthermore, the authors correctly include Trial as a random variable in their analysis (L221) but don’t include Rope as a nested random factor (each piece of algae is presented on a rope with other pieces and while it is reasonable that each rope is independent, the pieces on each rope are not). This nesting needs to be accounted for in the analysis. You should also include urchin ID as a random factor when an individual is used in multiple trials (L291). On a more minor note, it is nice to see the structure of your GLMM (L221) but you don’t need to tell us what you named your data frame.

More minor points:

1) L50. Surprised not to see something like Lessios HA. 2005. Diadema antillarum populations in Panama twenty years following mass mortality. Coral Reefs 24:125-127 as a citation

2) L52. Partly due – the Pacific has had an urchin mortality event

3) L60-61. Along with coral mortality that has opened up space.

4) L64-66. Recruitment of corals as an indirect and direct effect.

5) L86-87. See also Duran A, Adam TC, Palma L, Moreno S, Collado-Vides L, Burkepile DE. 2019. Feeding behavior in Caribbean surgeonfishes varies across fish size, algal abundance, and habitat characteristics. Marine Ecology 40:e12561.

6) L89-99. Key reference here is Burkepile DE, Hay ME. 2008. Herbivore species richness and feeding complementarity affect community structure and function on a coral reef. Proceedings of the National Academy of Sciences of the United States of America 105:16201-16206.

7) L106. Is it possible that feeding preferences have changed now compared to pre Diadema die-off because of the abundance of macroalgae?

8) L137. Worth noting somewhere that the nutritional content of Thalassia varies seasonally and may be more or less attractive to herbivores. Fourqurean JW, Escorcia SP, Anderson WT, Zieman JC. 2005. Spatial and seasonal variability in elemental content, δ13C-13, and δ15N of Thalassia testudinum from South Florida and its implications for ecosystem studies. Estuaries 28:447-461.

9) L143-144. Surprised no citation of Lewis SM. 1985. Herbivory on coral reefs: algal susceptibility to herbivorous fishes. Oecologia 65:370-375.

10) L170-174. Grazing varies over daily cycles so it would be worth highlighting in the Discussion that this could account for some of the deep / shallow differences.

11). L237-239. I’m not sure this totally solves the issue. If a fish goes straight for a particular species and eats it in one bite, doesn’t this still potentially represent a small percentage even though that species is clearly favored?

Reviewer 3 ·

Basic reporting

Addressing how grazing communities impact diverse assemblages of algae on degraded/recovering coral reefs has important implications for conservation management of these systems throughout the Caribbean. I believe the authors did a wonderful job communicating these points. The manuscript was very well written and I found the paper a very enjoyable and interesting to read. While I believe the authors have done a nice job introducing the project and relating and interpreting their findings back to the larger ecological concepts and applied implications, the paper does need some work on how they present their findings. I have a few comments that I have written below.

General/Major Comments:

• The written results section is detailed but I found it very hard to follow. Some of this isn’t a fault to the authors; they have many species of macrophytes and herbivores to describe and a reader who is not familiar with these species scientific names can get lost quickly. My suggestion is to (1) if possible, introduce the species (specifically fish) and use common names to describe them throughout the rest of text (e.g. Spotlight Parrotfish- Terminal). I realize that some of the macrophytes may not use a common name (for example Dictyota) but I was often confused when reading the results on which scientific name was an algae or a grazer. (2) -

• The figures could use a great deal of work. Here are my suggestions

o My overall suggestion which I think may help readers who may have less knowledge on the specifics of these macrophytes, is to organize the presentation of findings (within macrophytes) by defense mechanism. I could see this done in two ways
(1) have them ordered into defense mechanism groups on the X- or Y-axis in figures and
(2) color-code the data based on these mechanisms. The authors could have Figure 1, where the 10 macrophytes used are presented, grouped by the defense mechanism and potentiall draw a colored outline box around each grouping which corresponds to the color thoughtout the rest of the data presentation (e.g., Fig1 grouping 1 could be Chemical: dictyota, Laurencis sp 1 and 2, Lobophora, Galaxura and Halimeda- those could all be bordered with one color, while grouping 2 is structural. Galaxura and Halimeda might have two overlapping boarders to represent chemical and structural). For subsequent figures the three colors would be used to fill the bar charts or dot density plots. I believe, this would really help the reader interpret the authors figures better.

o Table 2 – this data works, but it could also be a nice figure (either a point or bar plot with corresponding error bars), where fish species is on the X-axis, Avg. # of fish on the Y-axis, and you could have two colors for deep vs shallow sites.

o Table 3 – I feel this is redundant with figure 4. If the authors feel this differently, please justify and/or potentially move to supplementary

o Table 4 – I also think this could be a figure as well where fish are on the X-axis and the y-axis has the ranked preferences (from unpreferred to highly preferred)

o Figure 2 – I am not convinced that this figure needs to be in the manuscript. I found it odd that the authors presented the data in this non-aggregated way (each trial) and feel it does not add anything to their findings and is quite redundant with the subsequent figure. Please justify if authors disagree and feel this presentation of the data is important.

o Figure 3 - I believe the authors were presenting A and B to show the macrophyte differences within a shallow or deep site and C is a test of macrophyte difference between sites? However, I found this confusing as I initially interpreted A and B as totally different data. My suggestion for this figure is to only present Fig3C and present statistical results in text or table.

o Figure 5 – I’m not sure why authors are presenting the data again by trial. Building in the mixing of treatments across trials should allow for aggregation of macrophyte species. If there is an interesting time component across trials, I could maybe see having an Fig5a and 5b for Dec 2017 and May 2018 but also think (if there were not differences in time) they should aggregate this data into one figure. If there are differences, authors need to explain why this might be.

Minor Comments

- Line 308:309 - Could the authors add justification to the text (and potential references if applicable) as to why they dropped at lower and upper grazing urchin trials

Experimental design

no comment

Validity of the findings

no comment

---

## Round 0.2 · accepted · Accept

Dear Dr. Spiers,

I want to express my heartfelt gratitude for the recent work made in the last version of your manuscript. I am delighted with your meticulous work and thrilled to see this manuscript moving to publication. I genuinely believe this paper can potentially become a seminal contribution to the field.

All the best

Juan Pablo

Reviewer 1 ·

Basic reporting

This version is a vast improvement from the last one. For the most part the English is clear, although there are some sentences that should be edited for clarity. For example, line 419-420 provides no information so does not add anything to the manuscript. There are multiple examples of sentences such as this in results section, so the whole section could be revised for clarity. (Some additional line comments are below.)

The background information is well explained, and the relevant literature cited. The introduction and abstract are well written and establish the background information nicely. The discussion does a good job of situating the study in the published literature. Lastly, the figures are appropriate to the data and are clear and well presented.

Experimental design

The questions addressed by this study (line 124-6) are not novel but are meaningful and thoroughly investigated with appropriate methodology. The methods section is much improved from the earlier version. It is clear, easy to read, and includes all the pertinent information.

Validity of the findings

The data are provided and support the conclusions drawn by the authors. Likewise, the figures reflect the data and findings well.

Additional comments

Line 102 ‘Makeup’ is confusing because it could mean community composition. Suggest using an alternative word.

Line 200 Not sure what you are referring to by ‘reef edge’. Do you mean reef crest? If not, then additional description would be useful.

Lines 386 and 411 are identical and provide next to no information. Suggest revising.

Figures 4 & 6 – this is a very trivial point, but instead of writing A and B in the two panels, you could write ‘Shallow’ and ‘Deep’. There is space in the panel, and it would allow the reader to see instantly what the figure is showing. Not an issue either way though.

Reviewer 2 ·

Basic reporting

The authors revisions have addressed my original concerns

Experimental design

The authors revisions have addressed my original concerns

Validity of the findings

The authors revisions have addressed my original concerns

Additional comments

N/A

Reviewer 3 ·

Basic reporting

I appreciate the authors edits to my and the other reviewers suggestions and believe that this manuscript is a lot more clear for me to follow and has important implications for coral reef grazer dynamics and overall ecosystem structure and function. I have no further comments or suggestions on my end. Thank you

Experimental design

no comment

Validity of the findings

no comment

Additional comments

no comment